# An Algorithm to Parallelise Parton Showers on a GPU

**Michael H. Seymour and Siddharth Sule**[*]

Department of Physics and Astronomy,
The University of Manchester, United Kingdom, M13 9PL

[*] siddharth.sule@manchester.ac.uk

## Abstract

The Single Instruction, Multiple Thread (SIMT) paradigm of GPU programming does not support the branching nature of a parton shower algorithm by definition. However, modern GPUs are designed to schedule threads with diverging processes independently, allowing them to handle such branches. With regular thread synchronisation and careful treatment of the individual steps, one can simulate a parton shower on a GPU. We present a Sudakov veto algorithm designed to simulate parton branching on multiple events in parallel. We also release a CUDA C++ program that generates matrix elements, showers partons and computes jet rates and event shapes for LEP at 91.2 GeV on a GPU. To benchmark its performance, we also provide a near-identical C++ program designed to simulate events serially on a CPU. While the consequences of branching are not absent, we demonstrate that a GPU can provide the throughput of a many-core CPU. As an example, we show that the time taken to shower $10^6$ events on one NVIDIA TESLA V100 GPU is equivalent to that of 295 Intel Xeon E5-2620 CPU cores.

# 1 Introduction

Monte Carlo Event Generators can accurately simulate high-energy physics and, hence, form a vital component of research at the LHC. That being said, they are computationally expensive: The ATLAS Detector's HL-LHC Roadmap document shows that event generators form around 17% of CPU usage [1]. This is because many simulated events are required to reduce the simulation uncertainty and allow exotic events (events with a very low probability of occurring) to be simulated. This document also states that even conservative CPU usage cannot maintain a sustainable budget. There is a demand for making event generators economical.

Two approaches have been presented to attain this requirement. The first involves profiling and finding bottlenecks in the current code, leading to immediate solutions [2] (see also [3] and other talks at the workshop [4]). The second involves adapting current event generation algorithms to run on a High-Performance Computer, which may contain multiple *Graphics Processing Units (GPUs)*. This is an active area of research, with recent publications, notably the PEPPER event generator [5] and the GPU version of MadGraph [6]. The Single Instruction Multiple Data (SIMD) or the Single Instruction Multiple Thread (SIMT) paradigm for GPU programming allows users to run repetitive tasks in parallel, increasing the throughput of the simulation [7]. This paradigm can be applied to event generation, as each event is independent. However, the GPU's requirement to execute the same instruction implies that threads cannot perform separate tasks. Hence, parton showers, which undergo different trajectories every time, are by construction not designed for GPU programming [1]. However, modern-day GPUs have the feature to handle branching code – the threads in the GPU can run more complicated, diverging tasks, and one can synchronise all threads at the end of the divergence [9]. We can use this feature, along with careful treatment of assigning tasks to threads, to simulate parton showers on a GPU.

We present a parallelised version of the Sudakov veto algorithm, capable of handling events in parallel. We also present a CUDA C++ implementation of the algorithm that simulates LEP events at 91.2 GeV at the partonic level and outputs jet rates and event shapes. We validate the program by studying the observables before comparing its execution time to a C++ program designed to simulate event generation on a single-core CPU. Although an "apples-to-apples" comparison cannot be made between simulating the shower on the CPU and the GPU, we work to be fair during our comparison.

We hope to make our work appealing to physicists and computer scientists alike. Hence, we provide brief introductions to both parton showers and GPU programming, but to avoid breaking up the flow of the paper for readers already familiar with those topics, we cover them in appendices: Appendix A and Appendix B respectively. References to further reading are also provided. The remainder of the paper is organised as follows. In Sect. 2, we describe the approach we take in implementing our parton shower algorithm in a form suitable for GPU running. In Sect. 3, we present and analyse the results, firstly briefly of the physics validation of our code, and then in more detail of its speed in comparison to the equivalent code run on a CPU, and an analysis of the associated energy cost. Finally, in Sect. 4, we make some concluding remarks.

# 2 The Parallelised Veto Algorithm

Today, most parton showers are simulated using the Sudakov Veto Algorithm [10]. In this algorithm, a generated emission at an evolution scale $t$ is accepted with a probability given

---

[1]Related issues were considered for a QED shower in [8]. Here, a partial synchronisation of threads was achieved by precalculating $n$-emission cases, leading to a speedup of 11 times overall.

by the ratio of the emission probability and its overestimate. If there are multiple possible emissions, the algorithm is run for all competing emissions and the one with the highest $t$ is deemed the *winner*. Figure 1 demonstrates this algorithm as a flowchart.

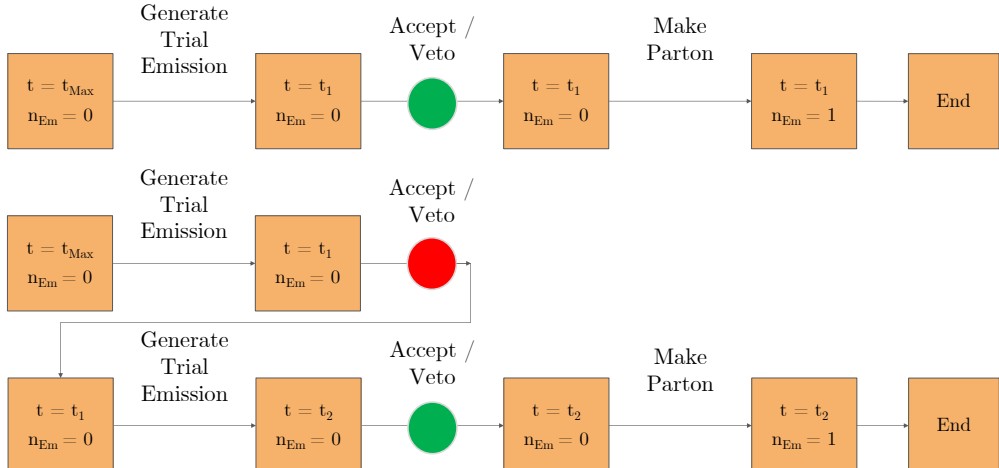

Figure 1: The Veto Algorithm, written as a flowchart. The boxes represent the state of the event, and the arrow represents a step of the process. There are two examples of possible routes here. In the first route, the check is successful (accept), and a new parton is generated at scale $t_1$. However, in the second route, the check fails (veto). This makes the code restart the while loop and generate the trial emission with argument $t_1$, giving $t_2$. The check is successful, and the parton is generated at $t_2$.

On a GPU, we make one fundamental change: each step of the veto algorithm is executed in parallel for all events. The algorithm is demonstrated in figure 2. For those interested in the GPU programming aspects of the algorithm, we provide pseudocode with comments in appendix C. This can be done because the steps of the algorithm are repeated in identical order, regardless of acceptance/vetoing of the emission (as seen by the symmetry in the columns in figure 1. The key change we make is that if the emission is vetoed in a given event, it is idle, while other events generate the new parton. Although this is inefficient, it ensures that all events are synchronised for the next step of the algorithm.

The central issue of the algorithm can be seen at the start of the second cycle. Generating a trial emission involves picking the highest possible emission from all particles in the system. This would involve iterating through the particles in the event, and the differences in the number of particles between events mean that each event wants to do something different from its surrounding events – leading to "divergence". This issue is further complicated by `if-else` statements within the loop (as a crude example, skip generating a trial emission if the particle does not have enough energy to emit). The divergence between events does not work in the SIMT paradigm, making parton showers unsuitable for GPUs by construction. However, modern GPU architecture contains functionality to allow divergence in the form of two features:

- **Warps:** GPUs split all their cores into batches, typically of 32, which run the same instructions simultaneously. Warps are entirely independent of other warps (like CPU cores) and managed by a warp scheduler. For example, a 256-thread GPU would behave like it has 8 "Cores", where each of them follows SIMT [11]. Hence, the problem of divergence between events is rescaled to 32 events.

- **Independent Thread Scheduling:** While the first implementations of GPUs assigned

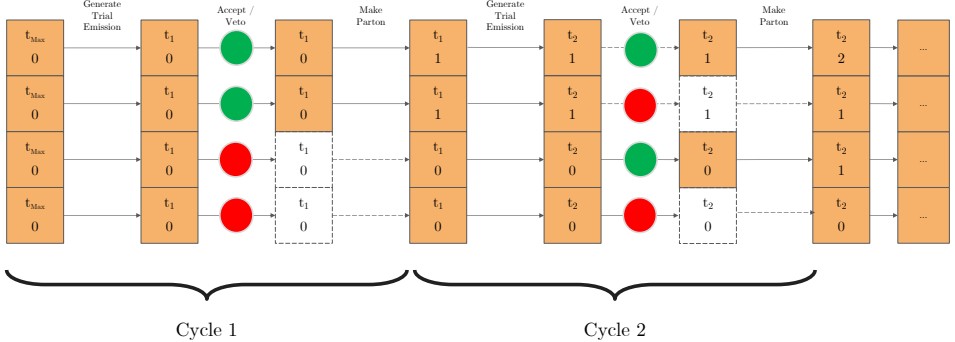

Figure 2: A flowchart for the GPU algorithm. Now, all events and their states are part of an array, and steps of the veto algorithm are executed in parallel for all events. Here, events **C** and **D** fail the first veto and do not make the new parton, while **B** and **D** fail the second veto and do not make a parton. This leads to all events being at the same stage but with different numbers of partons.

the same command for each core in a warp, modern GPUs allow each core to assign its own commands [9]. This allows each core to diverge from other cores to any extent. However, unlike CPU cores, different commands are interleaved. As an example, in the case where an `if-else` statement is provided to a warp, and 10 cores want to execute the `if` command and the rest want to execute the `else` command, the GPU would execute the `if` command for the 10 cores, while the rest are idle, followed by the `else` command while the first 10 cores are idle. Not only does the GPU allow the use of `for` loops and `if-else` statements, but it also automates them so that the user can write sophisticated, high-level code. The efficiency of the code is entirely dependent on the amount of divergence; optimal algorithms are designed to minimise branching (which is why we break down the veto algorithm into individual GPU steps).

We take advantage of these two features to implement the veto algorithm and parton shower on the GPU. In the next section, we study the impact of the features and the consequent inefficiencies on the time taken to execute events.

A final inefficiency encountered with this algorithm is when the showers reach the cutoff scale, $t_C$. For the serial approach, once an event reaches $t \leq t_C$, it stops, and the next event is showered. For the parallel approach, all events must reach $t \leq t_C$ for the shower to end, meaning that completed events must wait until all events are finished. Both of these cases are shown in figure 3. We haven't made any attempts to improve this here, and we will study its impact in the following section.

## 3   Implementation and Results for LEP at 91.2 GeV

We implemented the parallelised veto algorithm in a matrix element + parton shower + observables event generator in CUDA C++ for $e^+e^- \rightarrow q\bar{q}$. As a starting point, we used S. Höche's matrix element and dipole shower program from his "Introduction to Parton Showers" tutorial [12]. Although the tutorial and hence our implementation is adapted from the DIRE Shower [13], the parallelised veto algorithm can be applied to any parton shower model. We also implemented an event generator in C++ for a fair comparison with a CPU-only system. Apart from the CUDA syntax, the two generators are identical, demonstrating that one does not need to change the veto algorithm when showering on a GPU.

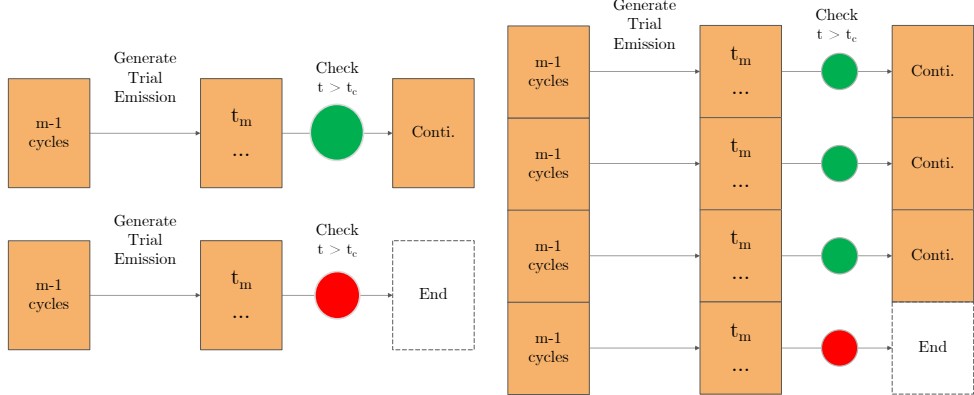

Figure 3: Flowcharts showing the serial and parallel veto algorithms' behaviour at the cutoff. As mentioned, the serial algorithm would start showering the next event, while the parallel algorithm would hold completed events until all events have finished showering.

We first validate the C++ and CUDA generators by replicating the results provided in Höche's tutorial. We then compare the execution times for the C++ program on a single core with the CUDA program for all stages of event generation. We also briefly comment on the power consumption of many-core CPUs and GPUs and discuss the implications of the observed execution times. We compared the two programs on:

- **CPU:** Intel Xeon CPU E5-2620 v4 @ 2.10GHz [14]

- **GPU:** NVIDIA Tesla V100 for PCIe, 16 GB [15]

## 3.1 Validation through Physical Results

The momenta of the final state partons were used to calculate jet rates using the Durham algorithm. The tutorial contained pre-calculated results of these distributions for validation purposes. As seen in figure 4, both the C++ and the CUDA generators agree with these results, confirming that the matrix element and parton showers have been correctly ported from the tutorial. The pre-calculated results come from a run of $10^5$ events, while our runs were of $10^6$ events each, which accounts for the difference in uncertainties, but it is clear that all three implementations agree within uncertainties. We have also checked that if the two implementations are provided with identical random numbers, they produce identical events (we plan to make this option available in a future version). We additionally provide results for the thrust and heavy jet mass distributions in figure 5. Although we do not have pre-calculated results to compare these to, the clear agreement is another test that the two implementations are identical. Both of these event shapes depend on calculating the thrust axis, which is notoriously computationally expensive but is ideally suited to GPU implementation – the heart of the computation, which is evaluated $\mathcal{O}(N^3)$ times for $N$ particles, just evaluates one dot-product and one two-way choice between one vector addition or subtraction. The programs store the distributions as Yoda files [16], which we plot using Rivet [17].

## 3.2 Comparison of Execution Times

We simulated a range of numbers of events up to $10^6$, as this was the maximum number of events the V100 GPU's memory could store. The simulations were run a hundred times, and

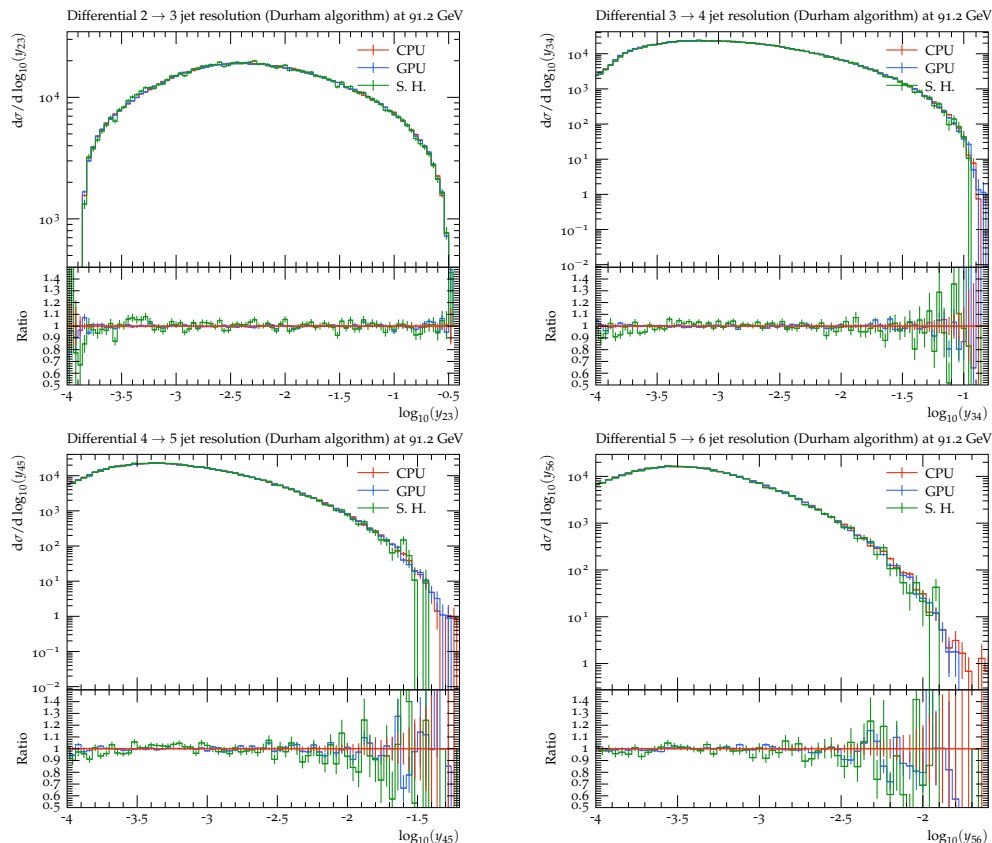

Figure 4: Jet Rates, using the Durham Algorithm. These observables are calculated using a jet clustering algorithm and are helpful to study the $p_T$ of emissions. All three showers demonstrate the same result, apart from some differences in random number generation. The C++ and CUDA results come from our CPU and GPU implementations, respectively, while those labelled S.H. are the results pre-calculated by Stefan Höche as part of his parton shower tutorial. The parameters used for the shower were $\alpha_s(m_Z) = 0.118$ and $t_C = 1$ GeV.

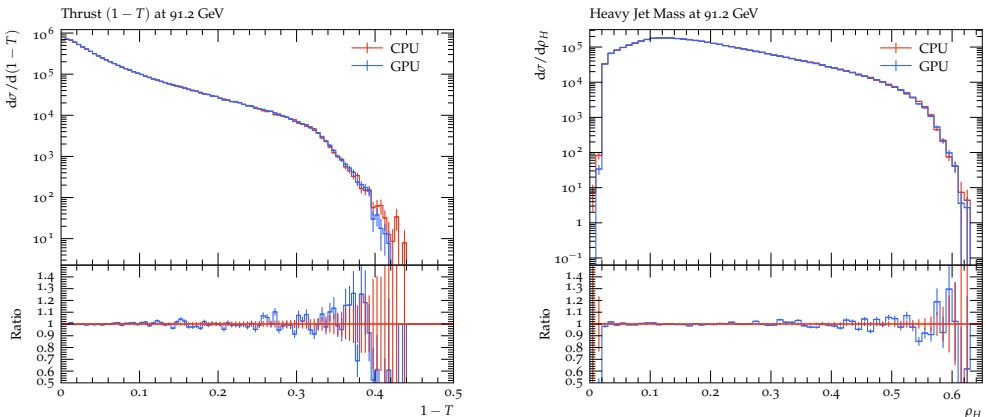

Figure 5: The Thrust and Heavy Jet Mass event shapes. These observables are helpful to study how the final state partons are distributed. For example, the Thrust distribution describes how "pencil-like" (closely distributed around an axis) an event is. The same simulation was used for these plots, and hence, the same parameters apply here.

the median and interquartile range were taken[2]. We used the `chrono` namespace of the C++ standard library to measure the real world ("wall clock") execution time taken at each step of the event generation – matrix element, parton shower and observables. Although there are CUDA namespaces available to measure just the GPU time, the CPU time must also be accommodated such that we compare the time taken for the *entire step*. That is, both the C++ and CUDA showers starting and ending at a common state. It is vital to mention that this is the closest possible approach to an 'apples-to-apples' comparison, which is not possible due to the varying architectures of the CPU and the GPU. The comparison of execution times is commonly used when comparing CPU-Only and CPU+GPU programs [18].

The execution times are shown in figure 6, arranged in the order of the event generation steps. Two features are consistently observed in all three stages. Firstly, the C++ generator is faster when simulating $\sim 1$ events, while the CUDA generator is faster when simulating $\sim 1,000$ events and more. This result is coherent with the properties of the CPU and the GPU. However, we also observe a steady increase in execution time on the GPU for $10,000$ events and more. This is a consequence of requesting more threads than cores on the GPU – the GPU has to distribute the events among the cores and handle them serially. The resulting impact on the execution time is reduced by the efficiency and latency hiding of the GPU [19]; the execution time of $10^6$ events is not a hundred times larger than the execution time of $10^4$ events. To illustrate this further, we applied a linear fit to the region of the increase, which confirmed that the gradient of the fits are always less than 1. At maximum capacity ($10^6$ events), the speedup achieved by the CUDA shower is 87 times for the Matrix element, 295 times for the parton shower, 182 times for observables and 275 times in total.

A curious feature is noticeable in the Matrix Element results at 2,000 events, which take approximately twice as long as 1,000 events, but this is not the start of a steady rise, which starts at around 20,000 events or more. We also found that in the region from 2,000 to 20,000 events, there was a fraction of events that took a lot longer than the average – while this fraction is small enough not to bias the average significantly, it does increase the standard deviation, which is why we preferred to show the interquartile range. Upon further study, it was discovered that allocating and freeing memory for the matrix element generator was the cause of this increase in both time and variability. These steps are done once per run and the memory they allocate is independent of the size of the events, yet their running time does increase with number of events. This is likely because the GPUs have on-board memories at various trade-offs of size and speed, which are automatically used as needed, depending on other memory usage for the events. These memory issues are more apparent in the Matrix Element step than the other two because its actual computation code is simpler and faster, exposing the memory moving times more clearly. This also highlights that the wall clock time is more complex and more relevant than just the sum of the computation times, as it provides a more realistic view of the simulation.

We also profiled the CUDA generator to split the total execution time by kernel (functions that are executed for every event) using NVIDIA's NSight Systems tool [22]. Table 1 shows the time consumed by the different kernels for one simulation of $10^6$ events. Selecting the trial emission takes the most time, as the kernel has to go through all possible pairs.

To further study the end of the parton shower, we studied the number of veto algorithm cycles taken to finish all events, for increasing number of events, shown in Figure 7. We observed that most events finished showering by 40-50 cycles. We also saw that the increase in a number of events directly corresponded to an increased waiting time at the end.

---

[2]The median and interquartile range were chosen, rather than the mean and standard deviation, due to a small fraction of events in which there were delays in the memory handling stages of event processing causing a small but significant tail, which we will discuss further below.

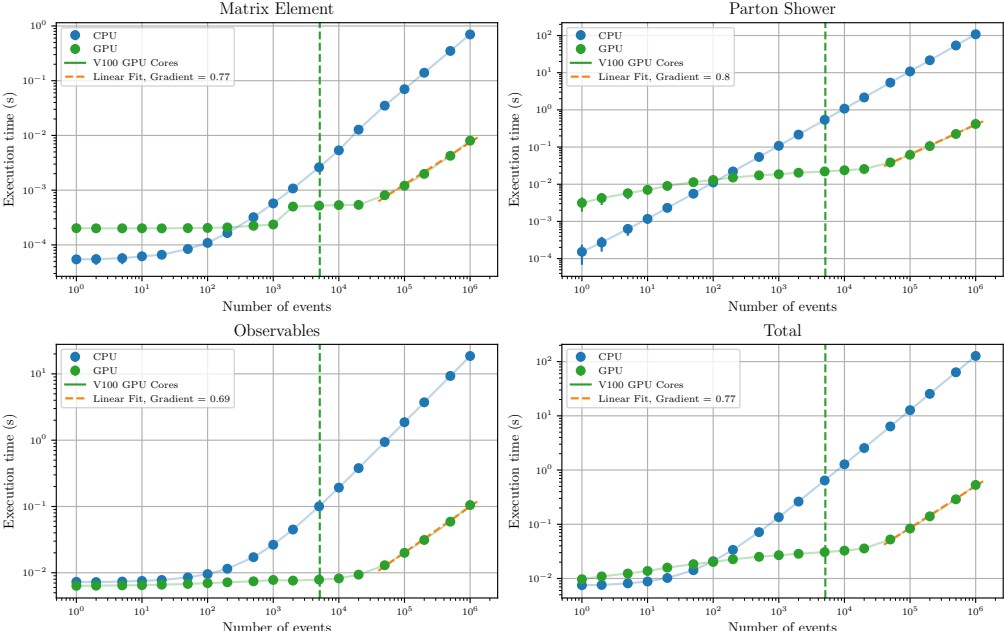

Figure 6: Execution times for the different event generation steps and the total event generation. As the matrix element is a leading-order analytical result, the function involved simple arithmetic and could easily be ported to CUDA. The parton shower benefits from the parallelised veto algorithm and independent thread scheduling. Not only are the observables calculated in parallel, their values are binned into histograms atomically, i.e. all at the same time (some examples of atomic histogramming can be found in [20,21]). The vertical line represents the number of cores in the V100 GPU. Beyond this, the GPU allocates waiting events to unoccupied warps. The linear fits on the GPU times in the steady-increase region have a gradient less than 1, which, on a log scale, implies a less-than-linear increase in execution time.

| Name | Instances | Total Time (ns) | Time (%) |
|---|---|---|---|
| Selecting the Trial Emission | 119 | 291,300,033 | 46.3 |
| *Device Prep.* | 1 | 105,578,119 | 16.8 |
| Vetoing Process | 119 | 45,518,957 | 7.2 |
| Thrust | 1 | 42,478,087 | 6.8 |
| Durham Algorithm | 1 | 26,657,439 | 4.2 |
| Checking Cutoff | 119 | 25,702,499 | 4.1 |
| Doing Parton Splitting | 119 | 23,646,846 | 3.8 |
| Calculating $\alpha_s$ | 119 | 20,654,227 | 3.3 |
| Histogramming | 1 | 17,950,803 | 2.9 |
| Matrix Element | 1 | 7,605,270 | 1.2 |
| *Set Up Random States* | 1 | 7,439,289 | 1.2 |
| Jet Mass/Broadening | 1 | 5,758,537 | 0.9 |
| *Validate Events* | 1 | 5,580,426 | 0.9 |
| *Prep Shower* | 1 | 2,651,686 | 0.4 |
| Set Up $\alpha_s$ Calculator | 1 | 8,800 | 0.0 |
| *Pre Writing* | 1 | 5,856 | 0.0 |
| Set Up ME Calculator | 1 | 3,968 | 0.0 |

Table 1: Statistics of the CUDA Kernels for a single run of $10^6$ events. The kernels in italics are built-in processes for managing memory and copying memory to and from the device. Device Preparation involves allocating memory for the event objects. Prewriting involves moving the histograms to the host (CPU).

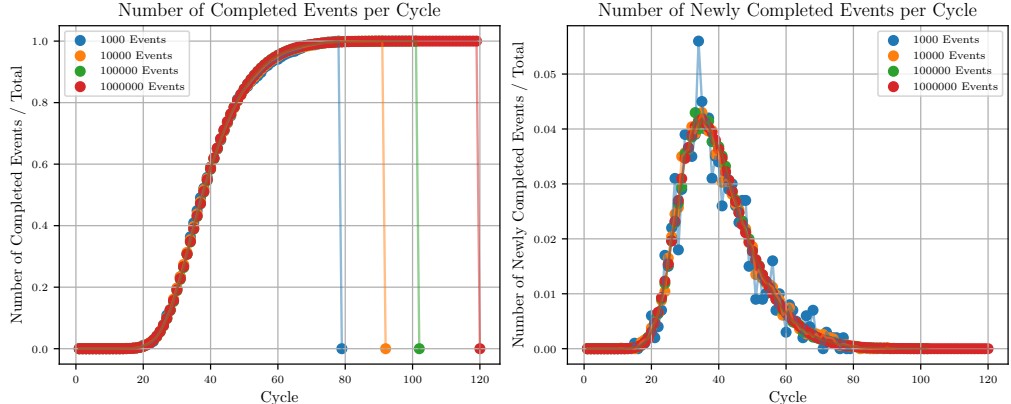

Figure 7: Number of completed events against the cycle, given as a cumulative and differential. It is important to mention that the number of cycles is not the same as the time – smaller number of events take a shorter time to complete a cycle. We also observed that near the end, when only a few events are active, the time taken to complete a cycle decreases. Hence, we believe using cycles instead of time works as a better unit for comparsion. In the cumulative plot, the vertical lines indicate the end of showering all events. One can see that the smaller event size leads to a shorter wait time. The differential plot shows that regardless of event size, most of the showers' events are finished in around 40 cycles. Note that this data is for an individual run, and the endpoint is subject to fluctuations. An interesting study could involve stopping the shower once a majority of the events have finished, and vetoing events where the shower scale is above the cutoff.

### 3.3 Comments on the cost of simulation

As our motivation is to make event generation sustainable, we connect our results with information on the devices' power consumption. The upper limit of the power consumed by a CPU chip is defined as the *Thermal Design Power (TDP)* [23, 24]. The CPU used for our tests, the Intel Xeon CPU, has a TDP of 85 W [14] and eight dual cores. This TDP value can be compared to the maximum power consumption provided by NVIDIA for the V100, which is 250 W [15]. Adding the TDP for one core [3] gives us a maximum consumption of 255 W. To match the performance of the GPU, one would need around 275 cores. Since the Intel Xeon has 16 cores in total, around 17 of them would be needed. This setup would consume 1445 W, five times more than the 1 CPU + 1 GPU setup (or equivalently, one Xeon core could be run for 17 times longer than the GPU, again costing five times more energy). Moreover, using 17 times as many machines or running 17 times longer would increase power overheads beyond that needed for our computation by 17 times.

## 4 Concluding Remarks and Outlook

In this paper, we demonstrate how the veto algorithm can be adapted to run on the GPU without changing its structure. In summary, this veto algorithm relies on running each step in parallel for all events. We also present a code that demonstrates a large throughput when compared to sequentially running the parton shower, even while experiencing the consequences of thread divergence. We hope this algorithm provides a starting point for more research on optimising GPU parton shower simulation while keeping the structure coherent to the original. One can intuitively change the splitting functions, colours, or kinematics.

From here, we plan to study whether we can obtain a similar speedup for a production-level parton shower. This would involve moving from a massless final state-only parton shower to a massive initial and final state parton shower, as seen in event generators like Pythia [25], Sherpa [26] and Herwig [27]. Fortunately, the veto algorithm is unchanged in this case; one must add the mass terms to the splitting functions and an extra step to evaluate Parton Distribution Functions for initial state showering. The PDF evaluation programs LHAPDF [5, 28] and PDFFlow [29] already offer multiple evaluations of PDFs on GPU.

The implementation of this algorithm, *GAPS* (a GPU-Amplified Parton Shower), can be found in the GitLab repository:

https://gitlab.com/siddharthsule/gaps

If you encounter any issues or want to discuss the CUDA C++ implementation further, please contact the corresponding author. This code will be public and maintained as an open-source project. Complete documentation and usage instructions are provided within the repository in the `doc` folder.

## Acknowledgements

The authors acknowledge using S. Höche's "Introduction to Parton Showers and Matching" tutorial. The authors would like to thank A. Valassi and J. Whitehead for comments on the preprint. The authors would like to thank the University of Manchester for access to the Noether Computer Cluster. SS would like to thank Z. Zhang for valuable discussions on CUDA programming, along with R. Frank for assistance with Cluster Computing.

---

[3] We profiled the application using NSight Systems again, and confirmed that only 1 Core of the 8-Core Xeon CPU is being used, and not the whole CPU.

**Funding information**    SS would like to thank the UK Science and Technology Facilities Council (STFC) for the studentship award. MS also acknowledges the support of STFC through grants ST/T001038/1 and ST/X00077X/1.

## A   Appendix: Parton Showers and the Veto Algorithm

Here, we summarise the motivation, fundamentals, and computational details of the parton shower. Parton showers have been in use for over forty years, and many texts have covered the topic thoroughly [12, 30–32].

For proton colliders like the LHC, QCD interactions are a significant component of the observed events. For an event generator to be considered "general-purpose", it must simulate the hard (high-energy, short-range) interactions of *partons*, a collective term for hadron constituents like quarks and gluons [30, ch. 4], and the soft (low-energy, long-range) interactions of hadrons, the physical bound states. However, perturbation theory can only be used in the hard regime; the soft regime is modelled using non-perturbative methods. As a solution, the factorisation theorem is used to separate and study these regimes independently [33] [30, ch. 7].

The two regimes are connected by the evolution of the *scale* (related to the momentum transfer in the interactions). This evolution occurs through the production of additional partons and the conversion of partons into hadrons. These processes are simulated in event generators using the *parton shower* and *hadronisation* models, respectively [31, ch. 1].

In a parton shower, quarks release energy by emitting gluons. These gluons split their energy when emitting further gluons or producing quark-antiquark pairs. These processes, termed *branchings*, lead to more partons with lower energies and smaller momenta. Consecutive branchings, like a quark emitting two gluons, occur at lower scales. Eventually, the scale of the branchings reaches the soft scale, where hadronisation models combine the final state partons into hadrons [30, ch. 5]. For example, parton showering in a typical scattering is shown in figure 8.

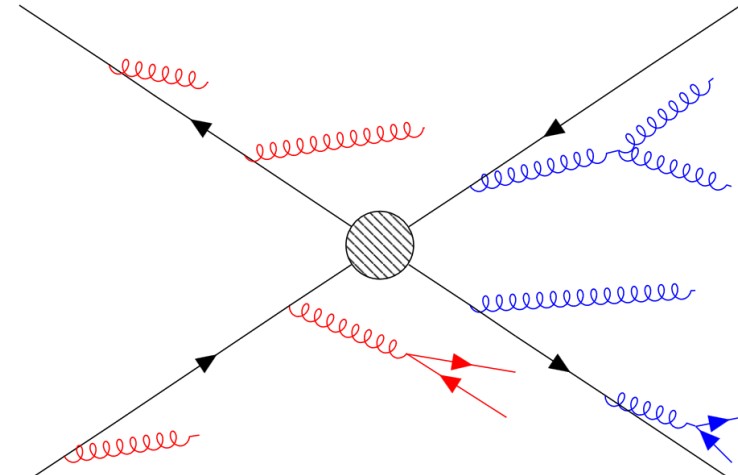

Figure 8: A fundamental interaction (black) with a parton shower in the initial state (red) and final state (blue). Assuming all the incoming and outgoing particles are quarks, they radiate gluons at high energies. For radiation before the interaction, one must accommodate the shower so that the quarks have the right amount of energy before interacting. This can be done by evolving backwards from the interaction and using PDFs [31].

Simulating the branching of a parton $\tilde{ij}$ to partons $i$ and $j$ involves generating a set of values $(t, z, \phi)$. The evolution variable $t$ defines the scale of the momentum transfer in the branching. The splitting variable $z$ defines how the energy from the splitter is divided between the children. The third variable, $\phi$, represents the azimuthal angle of the branching. Many variants of parton shower are possible, each with slightly differing definitions of $t$ and $z$, but all share these same general features.

To generate the distribution of $t$ values, we use the Sudakov form factor, defined as the probability that no emissions occur between the initial scale of the system $T$ and a smaller scale $t$ [34]. For this process, it is given by

$$\Delta_{\tilde{ij}\to i,j}(t,T) = \exp\left[-\int_t^T \frac{d\hat{t}}{\hat{t}} \int_{z_-}^{z_+} dz \, \frac{\alpha_s\left(p_\perp^2(\hat{t}, z)\right)}{2\pi} P_{\tilde{ij}\to i,j}(z)\right]. \tag{A.1}$$

Here, $z_\pm$ are limits on the $z$ integration, which are functions of $\hat{t}$ in general, and whose precise form depends on the precise definitions of $t$ and $z$, but which always obey $z_- > 0$ and $z_+ < 1$. $\alpha_s$ is the coupling strength of QCD, which, after renormalisation, can be considered a function of scale and, to reproduce a set of higher order corrections correctly, should be evaluated at a scale of order the transverse momentum of emitted gluons, $p_\perp$. $P_{\tilde{ij}\to i,j}$ is called the splitting function, derived from QCD in the collinear limit, which describes the probability distribution of the sharing of $\tilde{ij}$'s energy between $i$ and $j$. For gluon emission, i.e. when $i$ or $j$ is a gluon, $P_{\tilde{ij}\to i,j}(z)$ is divergent at $z = 0$ and/or $1$.

The Kinoshita-Lee-Neuenberg theorem shows that the Standard Model is infrared-safe: the divergences in virtual exchange and real emission integrals cancel each other [35–37]. Since the dominant (logarithmically-enhanced) finite parts of the real and virtual emission integrals are associated with these divergences, the corresponding probability distributions are unitary: the sum of the probabilities of these two types of emission sums to one. This is satisfied by the parton branching formalism; the virtual exchanges are accounted for in the no-branching probability, $\Delta$ and the real emissions are accounted for in $1 - \Delta$.

In Monte-Carlo sampling, $t$ is generated from $\Delta$ by solving

$$\Delta_{\tilde{ij}\to i,j} = \text{random}[0, 1]. \tag{A.2}$$

However, solving this is often not feasible, as the integrand

$$f(t) = \frac{1}{t} \int_{z_-}^{z_+} dz \, \frac{\alpha_s\left(p_\perp^2(t, z)\right)}{2\pi} P_{\tilde{ij}\to i,j}(z) \tag{A.3}$$

is too complicated to be analytically integrated and inverted.

In this case, an alternative method, called 'the veto algorithm', can be implemented [38]. Below, we provide a brief derivation based on [10, 34]. Here, a simplified form of the integrand is used, which must be greater than or equal to the current integrand[4] at all values of $t$. In our context, this 'overestimated' integrand $g$ is given by

$$g(t) = \frac{1}{t} \frac{\alpha_s^{\text{over}}}{2\pi} \int_{z_-^{\text{over}}}^{z_+^{\text{over}}} dz \, P_{\tilde{ij}\to i,j}^{\text{over}}(z) = \frac{1}{t}c, \tag{A.4}$$

where $\alpha_s^{\text{over}} = \alpha_s(p_\perp^2(t_C))$, $P_{\tilde{ij}\to i,j}^{\text{over}}$ is an overestimate of the splitting kernel and $z_\pm^{\text{over}}$ are constants satisfying $0 < z_-^{\text{over}} \le z_-$ and $1 > z_+^{\text{over}} \ge z_+$. These are defined such that $P_{\tilde{ij}\to i,j}^{\text{over}}$

---

[4]A simple extension to the algorithm also exists for cases in which the simplified form is not an overestimate, provided their ratio is bounded [39].

can be integrated analytically and inverted – needed for generating $z$. The collective term $c$ is a constant, and hence the indefinite integral of $g$ is

$$G(t) = c \ln(t) \longleftrightarrow G^{-1}(x) = \exp\left(\frac{x}{c}\right). \tag{A.5}$$

This result is then substituted into (A.2) to give

$$t = G^{-1}\big[G(T) + \ln(\text{random}[0,1])\big] = T \cdot \text{random}[0,1]^{\frac{1}{c}}. \tag{A.6}$$

To generate the corresponding value of $z$, the equation

$$\int_{z_-^{\text{over}}}^{z} \mathrm{d}z\, P_{\tilde{i}j \to i,j}^{\text{over}}(z) = \text{random}[0,1] \int_{z_-^{\text{over}}}^{z_+^{\text{over}}} \mathrm{d}z\, P_{\tilde{i}j \to i,j}^{\text{over}}(z) \tag{A.7}$$

can be solved similarly to give

$$z = I^{\text{over},-1}\big[I^{\text{over}}(z_-^{\text{over}}) + (I^{\text{over}}(z_+^{\text{over}}) - I^{\text{over}}(z_-^{\text{over}})) \cdot \text{random}[0,1]\big], \tag{A.8}$$

where $I^{\text{over}}(z) = \int \mathrm{d}z\, P_{\tilde{i}j \to i,j}^{\text{over}}(z)$. Assuming the azimuthal angle $\phi$ is uniformly distributed (false when considering spin correlations), a phase space point $(t, z, \phi)$ is generated. This phase space point is then accepted if

$$\text{random}[0,1] < \frac{\alpha_{\text{s}}\left(p_\perp^2(t,z)\right)}{\alpha_{\text{s}}^{\text{over}}} \frac{P_{\tilde{i}j \to i,j}(z)}{P_{\tilde{i}j \to i,j}^{\text{over}}(z)} \Theta(z_- < z < z_+) \tag{A.9}$$

This acceptance probability depends on the two components of $f$ that were changed to create the overestimate. The $\Theta$-function in Eq. (A.9) ensures that only $z$ values within the allowed range between $z_-$ and $z_+$ are accepted. If the point is accepted, the algorithm ends. If the point is rejected, or in other words, "vetoed", new values $(t', z')$ are generated from (A.6) and (A.8) with the substitution $T \to t$. This process is repeated until a new phase space point is accepted or until $t < t_C$, where $t_C$ is a fixed minimum scale, chosen to terminate the algorithm. The veto algorithm has been analytically proven to return the correct Sudakov form factor [10, p. 64].

To generate a subsequent emission after $(t, z, \phi)$, which we now rename as $(t_1, z_1, \phi_1)$, the veto algorithm is reimplemented with the Sudakov form factor $\Delta_{\tilde{i}j \to i,j}(t_2, t_1)$. In a parton shower, this process is repeated for all possible subsequent emissions for all the particles in the system. When multiple possible emissions exist, a trial emission of every kind is generated using the overestimate function and (A.2). The emission with the highest value of $t$ is deemed the *winner*, and $t$ is used as the proposed scale of the splitting. Since all emission types were "offered the chance" to emit at scales above $t$, it is used as the upper scale for the subsequent evolution of all partons, not only the products of the generated splitting.

In modern event generators, two classes of parton shower algorithms can be distinguished: in the first, the parton shower is developed for each parton individually as a series of $1 \to 2$ branchings. Energy and momentum cannot then be conserved because the sum of the momenta produced in a branching has an invariant mass that is greater than the parent's. This is rectified by a final stage of the algorithm, in which small amounts of energy and momentum are shuffled between partons to ensure that they are conserved. In the second class, often called a *dipole shower*, the emission from a parton is generated with reference to one or more additional partons, with which energy and momentum are shuffled immediately so that they are conserved after each branching. This is sometimes characterised as a $2 \to 3$ branching, but

might more properly called $1(+n) \rightarrow 2(+n)$ with $n \geq 1$, since it is properly a $1 \rightarrow 2$ branching in the presence of $n$ "spectator" partons [40]. To illustrate our discussion of GPU algorithms, we have implemented the simplest possible dipole shower with a single spectator called the colour partner.

The pseudocode algorithm below explains how the parton shower runs and is written using S. Höche's tutorial [12]. This algorithm is also shown as a flow chart in figure 1.

The heart of the algorithm is the function `SelectWinnerEmission`, which loops over partons, offering each the chance to emit. It finds the generated emission with the highest scale and returns it, provided that this is higher than the minimum allowed scale `t_C`. It assumes that information about the partons in the event and the current value of $t$ are available as global variables.

```
function SelectWinnerEmission:

    winner_scale = t_C
    winner_splitting_function = None

    # For a Dipole Shower, we try all kernels for all splitter
    # spectator combinations, and see which one generates
    # the highest t
    #
    # In our CPU Shower, we can define the parton
    # list as an empty array, where new partons are appended
    # to the array as emissions are generated (this is not
    # quite the same in the GPU case)
    for splitter in parton_list:
        for spectator in parton_list:

            # Ensure that splitter != spectator
            if splitter = spectator:
                ignore

            # Leading Colour -> colour connected dipoles
            if splitter, spectator != color_connected:
                ignore

            for splitting_function in split_funcs

                # P(u -> ug) may not be same as P(d -> dg), etc.
                if splitting_function.splitter != splitter:
                    ignore

                temp_scale = splitting_function.choose_scale()

                if temp_scale > winner_scale:

                    winner_scale = temp_scale
                    winner_splitting_function = splitting function

    # Winner Emission Found!
    return winner_scale, winner_splitting_function
```

The function `GenerateEmission` uses `SelectWinnerEmission` repeatedly to select a winner emission, calculates the veto probability for this emission and, if accepted, reconstructs the momenta of the produced partons and spectator.

```
function GenerateEmission:

    while t > t_C:

        # Generate Emissions and Determine Winner - above
```

```
        winner_scale , winner_splitting_function = SelectWinnerEmission
            ()

        t = winner_scale

        # To ensure we don 't make a new parton under the cutoff
        if t > t_C:

            # Veto: A vital step in the Parton Shower
            p = generate_veto_probability ()
            if random [0 ,1] < p:

                # Do Physics
                solve_kinematics ( splitter , spectator , t, z)
                assign_colours ( splitter , winner_kernel )

                # Change the momenta and colour of current partons
                update_emitter_and_spectator ()

                # Add new parton to system ( Emitted )
                new parton ( momentum , colour , ...)

                # Ends Function after Branching is Done
                break
```

Finally, a very simple main program can use these to shower quark-antiquark events at a fixed centre-of-mass energy:

```
# Start the Shower by setting the starting t
parton_list = [quark , antiquark ]
t = t_max = CoM_Energy ()

# A while loop repeatedly calls GenerateEmission
# until the system is full of partons and t = t_C
while t > t_C
    GenerateEmission ()
```

# B  Appendix: An Introduction to GPUs and GPU Programming

In this section, we highlight the key elements of GPUs and how to adjust C++ code to utilise them. This section provides context for our new algorithm and provides further reasoning as to why we cannot do an "apples-to-apples" comparison between the CPU-only and CPU+GPU showers. The seminal text on this topic is the NVIDIA CUDA C++ Programming Guide, which is regularly updated alongside new releases of GPUs and updates to the language [11]. Additionally, one can further look into the *GPU Computing Gems* series of books, which contain techniques and examples relevant to scientific computing [41].

The majority of computers are built following the Von Neumann Architecture, where a Central Processing Unit (CPU) is in charge of undertaking complicated calculations, managing memory and connecting input and output devices [42]. Some computers also come with multiprocessors, or multiple *CPU cores*, allowing one to parallelise tasks or alternatively compute different tasks [43]. For this reason, data centres worldwide provide computer clusters with hundreds of cores. However, suppose the task is simple and can easily be parallelised. In that case, it might also be beneficial to compute it on a *Graphics Processing Unit (GPU)*, which contains thousands of smaller, less powerful cores that are managed as one. This architecture is designed such that all cores execute the same command at a given time, which makes it a valuable tool for solving simple *embarrassingly parallel* problems, where little effort is needed

to parallelise the task [44]. For example, if the elements of two arrays are independent, a CPU would add them one at a time, but a GPU would add every element in parallel. That being said, the smaller cores in the GPU may not be as fast for sophisticated tasks, so it is important to utilise both CPU and GPU when computing. This is known as heterogeneous programming [11]. Figure 9 summarises the difference between CPU and GPU cores.

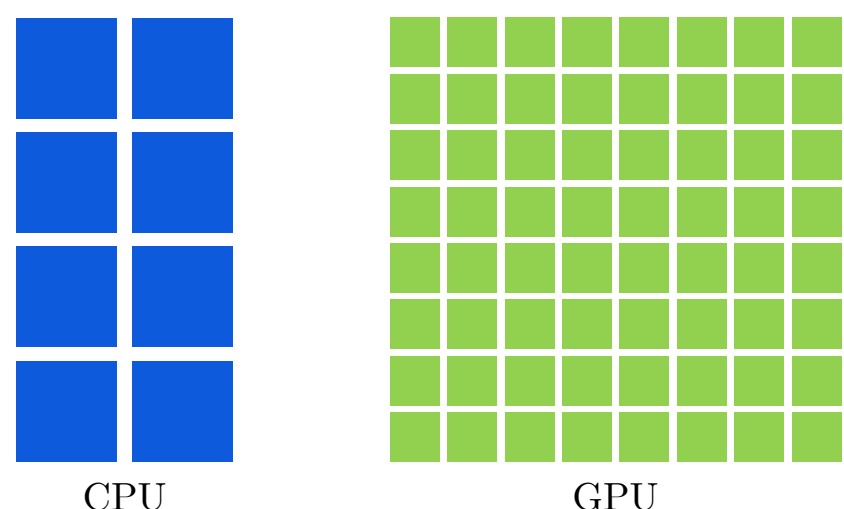

CPU                                            GPU

Figure 9: CPU and GPU cores. Here, the size of the core is used as a reference for its computing ability. As mentioned, the CPU has a small number of powerful cores, while the GPU has thousands of less powerful cores. This difference makes them suited for different tasks.

Programming on a GPU can be done using languages such as CUDA, HIP [45], OpenCL [46] or Kokkos [47]. We use CUDA (C++ version) in our program, which is specifically designed for the NVIDIA GPUs. Programming on a GPU is similar to regular programming but involves two crucial components: distributing tasks on the GPU cores and moving information from the CPU (often called *host*) to the GPU (often called *device*) and back. We provide a few beginner-friendly CUDA examples in the doc folder of the *GAPS* repository. As an advanced example, we look at matrix element generation here. For a leading order process like $e^+e^- \rightarrow q\bar{q}$, the solution is known. In our code, we randomly generate a flavour, compute the matrix element, and then use it to calculate the differential cross section. For simplicity, we neglect data related to the kinematics of the event from our example. In a CPU-only program, one would use a for loop to generate one random number, calculate the matrix element, and calculate the differential cross section. On a GPU, this for loop can be replaced by a "kernel" (not to be confused with splitting kernel!), which parallelises this task:

```
# Device = Function that can only run on the GPU
# If on GPU, can be called by all threads at once
__device__ function MatrixElement(int flavour)
    # Formula Goes Here

# Global = Operates from the CPU and Runs on the GPU
# This is how you make each thread calculate one ME and XS
__global__ function calcDifferentialCrossSec(double *xs_data, int N)

    # Pseudocode for getting Thread ID
```

```
    idx = getThreadID()

    # Safety Check:
    # Don't run if idx is greater than number of needed events
    if (idx >= N): return

    # Get random number for flavour
    flavour = cuda.random(1, 5)

    # Calc ME and XS
    ME = MatrixElement(flavour);
    xs = # Formula to convert ME to XS

    # Set the value
    xs_data[idx] = xs

# Function Run Here

# Number of Events
N = 10000

# Make arrays on CPU and on GPU
double *host_xs, *device_xs;
malloc(host_xs, sizeof(double) * N)
cudaMalloc(device_xs, sizeof(double) * N)

# Launch a Kernel of size N for N events
kernel<N> calcDifferentialCrossSection(device_xs, N)

# Copy info on the device to the host
# Because we cannot write or store from the device
# Do this as few times as possible as it is very
# time/memory-consuming
#
# NB: We often have to mention the direction in
# which the memory is being copied. Here we
# specify copying from Device to Host
cudaMemcpy(host_xs, device_xs, DeviceToHost)
```

This way, the matrix elements can be calculated for many events without needing a many-core CPU. The function `MatrixElement` is usually complicated enough that it takes longer to evaluate once on a GPU than on a CPU, but this is more than compensated by the fact that it can be calculated hundreds or thousands of times in parallel on the GPU.

## C   Appendix: Pseudocode for the GPU Parton Shower

In the GPU Implementation, the steps are written as CUDA Kernels instead of functions. The following kernels were used in our algorithm:

- Selecting the Winner Emission: Notice that almost all of it is identical to the CPU version in Appendix A. Apart from being a kernel rather than a function, the only difference is the simple flag that leaves the kernel very quickly if the shower has already terminated. This is extremely important for the efficiency of our implementation.

```
__global__ function SelectWinnerEmission(object *events, int N)

    int idx = getThreadID();

    if idx >= N
```

```
        return

    # This is a VERY Important step
    # If the shower has ended, the GPU Core will be assigned
    # a different event. This is why the code is so fast, and
    # why doing more events than GPU cores is better. We set
    # this parameter after getting the new scale t.
    #
    # Note: This is set to true in the CheckCutOff Kernel Below
    if events[idx].endShower = True
        return

    winner_scale = t_C
    winner_splitting_function = None

    # This is the EXACT SAME code as the simple shower
    # that is provided in appendix A. We consider this
    # as the biggest success of this algorithm. We
    # remove the informative comments from that version
    # here
    #
    # IMPORTANT: While we can reallocate memory on the
    # GPU, it is very time consuming. Hence, instead of
    # appending new elements to a dynamic-sized list,
    # we preallocate a set number of partons to every event
    # (this is set to 50 for LEP, but in the future we can
    # go higher, at the cost of fewer events being executed
    # in parallel. For fair comparison, we also use this
    # method in the CPU Shower)
    for splitter in parton_list:
        for spectator in parton_list:

            if splitter = spectator:
                ignore

            if splitter, spectator != color_connected:
                ignore

            for splitting_function in split_funcs

                if splitting_function.splitter != splitter:
                    ignore

                temp_scale = splitting_function.choose_scale()

                if temp_scale > winner_scale:

                    winner_scale = temp_scale
                    winner_splitting_function = splitting function

    event[idx].winner_splitting_function =
        winner_splitting_function
    event[idx].t = winner_scale
```

- Checking the Cutoff, $t > t_C$

```
__global__ function CheckCutoff(object *events, int N)

    int idx = getThreadID();

    if idx >= N
        return
```

```
if events[idx].endShower = True
    return

if event[idx].t <= t_C

    # The Default value for all events is false.
    # Once the Cutoff is reached, this is set to
    # True.
    #
    # Before doing anything, the code checks if the
    # event has ended. If it has, nothing
    # is done in that thread.
    event[idx].endShower = True

    # Add to the completed counter
    # Atomic = multiple threads at the same time
    AtomicAdd(completedEventsCounter, 1)
```

- Acceptance/Vetoing Procedure

```
__global__ function AcceptOrVeto(object *events, int N)

    int idx = getThreadID();

    if idx >= N
        return

    if events[idx].endShower = True
        return

    p = generate_veto_probability(event[idx])
    if random[0,1] < p
        event[idx].acceptEmission = True
    else
        event[idx].acceptEmission = False
```

- Generating the Splitting for Accepted Emissions

```
__global__ function GenerateEmission(object *events, int N)

    int idx = getThreadID();

    if idx >= N
        return

    if events[idx].endShower = True
        return

    if event[idx].veto == True

        solve_kinematics(splitter, spectator, t, z)
        assign_colours(splitter, winner_kernel)
        update_emitter_and_spectator()
        new parton(momentum, colour, ...)
```

These kernels are called using a host (CPU) Function

```
function runShower(N)

    object events = FixedOrderEvent(N)
```

```
while completedEventsCounter < N

    kernel<N> SelectWinnerEmission(events, N)
    kernel<N> CheckCutoff(events, N)
    kernel<N> AcceptOrVeto(events, N)
    kernel<N> GenerateEmission(events, N)

# Now you have an event with Hard and Soft Particles.
```

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
