# Peer review of "An Algorithm to Parallelise Parton Showers on a GPU"

_SciPost Physics Codebases, doi:SciPost Phys. Codebases 33 (2024) , SciPost Phys. Codebases 33-r1.1 (2024)_

## Round 2 · Referee Report · Pierre-Antoine Delsart (Referee 2) · 2024-5-27

Strengths

1- demonstration of the feasability of a parton shower algorithms running on GPU 2- Clear explanations for a wide audience

Report

The article describes a new implementation of a particle collision generator algorithm which runs on Graphical Processing Unit (GPU) computer system. Crucially, the algorithm includes the simulation of the parton shower (the QCD phenomena resulting from the emission of a quark of a gluon in the collision), by implementing a GPU version of the "Sudakov veto algorithm". As clearly exposed and demonstrated in the article, the main advantage of this algorithm is its massively parallel execution on GPU, resulting in much faster or much more power efficient calculations than the same algorithm run on regular CPU. The paper is well written and provide clear explanations, in particular through the annexes in which additional useful details for non experts are given. The methodology is simple and robust and in my opinion supports the conclusions of the authors. The article tackles an important problem for the future of particle physics research : obtaining a sustainable way of computing the necessary simulations to exploit the HL-LHC or potential future collider experiments. As the authors provide a public repository for the source code of their algorithms, the reported work represents a concrete proof-of-concept for a solution of this problem in the event generation domain. In my opinion a missing point is a discussion on how this work relates practically with existing generator software (Pythia, Herwig, others...), in particular if this implementation is compatible with others and/or can be easily ported (there's a sentence suggesting it is the case, but the point is to develop on this) and if it can be extended to more complex evaluations including higher-order calculations. Such a discussion would be more of an opinion rather than a scientific demonstration, but honest insights on how the algorithms can be generalized or adopted by the community would extremely instructive for all readers.

As a summary, I think this work is absolutely worth publishing, but I would insist the above discussion points are added, probably in the conclusion of the article. Beside, I have other non-blocking minor questions and suggestion to clarify the document below.

  • Figure 2 & annexe : For non experts, it is not immediately obvious what exactly is calculated as part of one parallel flow ( i.e. horizontal arrow connected boxes) : is it the successive emissions from 1 parton or all the emissions in one collision event. Simply stating it explicitely, ex as part of the caption of fig 2, would ease the reading.
  • Figure 4 : the variables are not described. It would be good to give a description, even if very brief and an explanation of why they are relevant w.r.t. simpler variable such as pT or rapidity.
  • figure 7 : Is the following interpretation correct : at cycle=80 we have 95% of events done but we need to wait 40 more cycle (That is 50% time more) for the parton shower step to end and move to observable calculation ? If so, is this (=waiting 50% longer for 5% of events) an intrinsic limitation of using GPUs ? if so this is an interesting point I would suggest is made in the article.
  • Annexes : the pseudo-code are generally useful as they convey well the general ideas of the algs. There are a few confusing points which could be clarified :
    • the is the parton_list in the 1st example a dynamic list owned by each thread ? Is it populated by the line "new parton(...)" in the 2nd example ?
    • what is the "DeviceToHost" at the end of annex B ?
    • there is multiple reference to "events [ idx ]. endShower" but it is nt very clear at which point in the examples this could be set to True ?

Requested changes

1- Discussion on how this work relates practically with existing generator software (Pythia, Herwig, others...), in particular if this implementation is compatible with others and/or can be easily ported (there's a sentence suggesting it is the case, but the point is to develop on this) and if it can be extended to more complex evaluations

Recommendation

Ask for minor revision

  • validity: top
  • significance: good
  • originality: ok
  • clarity: high
  • formatting: good
  • grammar: excellent

Author:  Siddharth Sule  on 2024-05-31  [id 4531]

(in reply to Report 2 by Pierre-Antoine Delsart on 2024-05-27)
Category:
answer to question

Dear Dr. Delsart, thank you for the kind feedback! We'll take it into account as we work on the new draft of the paper.

You have raised an important point about the implementation of parton showers in general-purpose event generators and whether they will benefit from the same speedup we see. As a first development towards parton showering on GPUs, we feel that a full production-ready implementation is beyond our current scope. Nevertheless, we have considered the additional steps that would be needed for a general implementation, in particular, the inclusion of parton masses and of initial-state radiation, which necessitates an evaluation of parton distributions at every step, both of which affect the physics formulae, but not fundamentally the algorithm. Production-ready parton showers define the physics of initial and final state emissions and prove their validity by simulating LHC, DIS and LEP events [arXiv:0709.1027], [arXiv:0909.5593]. In our work, we have only studied LEP, and our current research aims to simulate LHC and DIS events. The only significant modification to the veto algorithm is an extra step to evaluate a ratio of parton distribution functions on the GPU (demonstrated in [arXiv:2311.06198]). We plan to construct C++ and CUDA showers, connect our code to the PDF program and repeat our validation and comparison for LHC and DIS purposes. While there are issues to be addressed, we do not see any "showstoppers".

In terms of this paper, we will add a paragraph to address these comments and resubmit it in the coming weeks. We will mention the page and section where we've added these updates, too.

Thank you for the minor comments - we will update the figures and pseudocode based on them.

Once again, thank you for the feedback!
The Authors

---

## Round 2 · Referee Report · Anonymous (Referee 1) · 2024-5-27

Report

This paper proposes an algorithm for computing parton showers on a GPU.
It is therefore at the interface of physics and computer science.

The challenge is well motivated: simulation of parton showers is very resource intensive and presents a major bottleneck for experimental collaborations and theorists alike, while GPU hardware is becoming increasingly widespread and powerful, with the possibility of executing vast number of instructions simultaneously.

The paper is very clearly written, with a discussion of GPU architecture and the differences between programming for CPUs and GPUs; the algorithm is clearly described in terms of a flowchart and pseudocode in the appendix; there is an implementation with a physical application, and clear comparison of GPU and CPU performance; the implementation code is made public; the result is that their algorithm is much faster than the CPU equivalent and uses less energy. The paper is easy to read and interesting.

On the other hand, the idea behind their algorithm appears essentially just run each event on one GPU core, and run events in parallel. There is therefore little that is particular to a GPU compared to CPU computing, only that GPUs typically have of order 100 times as many cores as a CPU, and the authors admit that modern GPUs are designed with enough freedom that "allows developers to write sophisticated algorithms with little concern for optimisation or thread usage". Originally GPUs were designed to run many copies of simple commands, and therefore could not handle multiple branchings. This meant that parton showers were by default not suited to GPUs; the authors have not solved this problem, but instead rely on newer GPU technology being better able to handle warp divergence. One concern that I have is that, since their algorithm is essentially a toy model, an extrapolation to a full parton shower code would see the benefits reduced or disappear. I would appreciate if the authors could add some comments on this and the future prospects (of which currently there are none beyond some hopeful words), but otherwise in light of my above positive remarks I recommend the paper for publication.

Requested changes

  1. Comment on the possibility of extrapolating to a more advanced parton shower algorithm and whether the GPU benefits will remain.

Recommendation

Ask for minor revision

  • validity: -
  • significance: -
  • originality: -
  • clarity: -
  • formatting: -
  • grammar: -

Author:  Siddharth Sule  on 2024-05-31  [id 4530]

(in reply to Report 1 on 2024-05-27)
Category:
answer to question

Dear Referee, Thank you for your kind words and insightful comments! We'll take them into account as we update the paper.

Regarding your comment on the algorithm's originality, we would like to emphasize that our approach is indeed unique. Unlike a 100-core CPU, our code doesn't run a hundred parton showers in parallel. Instead, we convert each algorithm step into a cuda kernel, enabling the evolution of a hundred states to occur simultaneously, regardless of parton multiplicity. This innovative method optimizes tasks like evaluating alpha_s and checking the cutoff, all of which are done simultaneously, contributing to a significant speedup. We understand that we need to elaborate on this aspect, and we will do so in the next draft of the paper.

Regarding your concerns about the model's simplicity, we'd like to clarify that the Parton Shower Tutorial, which our code is based on, is a fully functional final state shower. It operates under the assumption that the quarks are massless. Any inclusion of quark masses would require adjustments to the physical formulae, not the algorithm itself. That being said, your concern still stands—in this work, we have not considered emissions from the initial state, issues related to the break-up of the incoming hadrons, or hadronization. These are all interesting issues of our current and future work. Production-ready parton showers define the physics of initial and final state emissions and prove their validity by simulating LHC, DIS and LEP events [arXiv:0709.1027], [arXiv:0909.5593]. In our work, we have only studied LEP, and our current research aims to simulate LHC and DIS events. The only significant modification to the veto algorithm is an extra step to evaluate a ratio of parton distribution functions on the GPU (demonstrated in [arXiv:2311.06198]). We plan to construct C++ and CUDA showers, connect our code to the PDF program and repeat our validation and comparison for LHC and DIS purposes. While there are issues to be addressed, we do not see any "showstoppers".

In terms of this paper, we will add a paragraph to address these comments and resubmit it in the coming weeks. We will mention the page and section where we've added these updates, too.

Once again, thank you for the feedback!
The Authors

---

## Round 3 · Referee Report · Anonymous (Referee 1) · 2024-7-18

Report

I thank the authors for their wholehearted response to my report: they implemented my requests and I am very happy to recommend publication of the paper in its current form.

Recommendation

Publish (easily meets expectations and criteria for this Journal; among top 50%)

---

## Round 3 · Referee Report · Pierre-Antoine Delsart (Referee 2) · 2024-7-30

Report

I thank the authors for their detail answers to the questions and remarks from my 1st report.
I have reviewed the revised their version which adresses the points that were made and improve the quality of the document.
Therefore I have no more objection to the publication of this work.

Recommendation

Publish (easily meets expectations and criteria for this Journal; among top 50%)

---

## Round 3 · Author Response

Dear Editor and Referees,

Apologies for the delayed update, but we have now submitted the updated paper. We have addressed the referees' queries and concerns (see list of changes). We hope you find these changes suitable for your concerns. Please don't hesitate to contact us for further clarifications/concerns.

Thanks!
The Authors

---

## Round 3 · List of Changes

1. While making the revisions, we discovered a minor physics bug in the code. After fixing the bug, we observed minor changes to the speedups (the Parton Shower Speedup went from 260 to 295). This has NOT affected any of our findings or theory; it is just a change in the number in the manuscript.

  2. Revamped the Algorithm Section: After learning more about GPUs over the last few months, we have written a more accurate explanation of why the algorithm works and how it differs from a multi-core simulation (based on the feedback by referee 1). We hope that it now clearly explains why this is a novel approach. To further avoid any confusion, we've removed the use of the word 'simultaneously' and replaced it with the more coherent 'in parallel'.

  3. We Added comments about future implementation and relevance to production-ready showers in the conclusion section.

  4. Added a few references - One Paper in the intro ([2]) and GPU Histogramming References

  5. Updated the Execution Time Slide - Added vertical line where the number of events = number of cores of the GPU, which helps explain the increase in runtime

  6. Addressed Comments by Dr. Delsart:

  7. Figure 2 & annexe : For non experts, it is not immediately obvious what exactly is calculated as part of one parallel flow ( i.e. horizontal arrow connected boxes) : is it the successive emissions from 1 parton or all the emissions in one collision event. Simply stating it explicitely, ex as part of the caption of fig 2, would ease the reading. - ADDED A FEW SENTENCES TO FIGURE 2, AS WELL AS TO FIGURE 1 TO EXPLAIN BETTER

  8. Figure 4 : the variables are not described. It would be good to give a description, even if very brief and an explanation of why they are relevant w.r.t. simpler variable such as pT or rapidity. - DONE

  9. figure 7 : Is the following interpretation correct : at cycle=80 we have 95% of events done but we need to wait 40 more cycle (That is 50% time more) for the parton shower step to end and move to observable calculation ? If so, is this (=waiting 50% longer for 5% of events) an intrinsic limitation of using GPUs ? if so this is an interesting point I would suggest is made in the article. - ADDED COMMENTS EXPLAINING CYCLE != TIME, AND TALKED MORE ABOUT THEIR RELATION

  10. Annexes : the pseudo-code are generally useful as they convey well the general ideas of the algs. There are a few confusing points which could be clarified :

  11. the is the parton_list in the 1st example a dynamic list owned by each thread ? Is it populated by the line "new parton(...)" in the 2nd example ? - ADDED COMMENTS TO CODE

  12. what is the "DeviceToHost" at the end of annex B ? - ADDED COMMENTS TO EXPLAIN THIS

  13. there is multiple reference to "events [ idx ]. endShower" but it is nt very clear at which point in the examples this could be set to True ? - ADDED COMMENTS IN A FEW SECTIONS TO EXPLAIN

---

## Editorial Decision

published